# A Combined Positioning Method Used for Identification of Concrete Cracks

**DOI:** 10.3390/mi12121479

**Published:** 2021-11-29

**Authors:** Jianzhi Li, Bohao Shen, Junjie Wang

**Affiliations:** 1Structure Health Monitoring and Control Institute, Shijiazhuang Tiedao University, Shijiazhuang 050043, China; 2School of Mechanical Engineering, Shijiazhuang Tiedao University, Shijiazhuang 050043, China; shenbohao0105@163.com; 3School of Materials, Shijiazhuang Tiedao University, Shijiazhuang 050043, China; junjiewang315@163.com

**Keywords:** localization, fiber Bragg grating, fully distributed fiber optic sensors, concrete beam

## Abstract

Fully distributed fiber optic sensors are characteristically used for the measurement of long distances and continuous distribution of space. However, due to the different fiber type, fiber length, ambient temperature and strain, fully distributed fiber optic sensors fail to locate damage accurately and cause a greater error. Therefore, this paper proposes a new positioning method of combining fully distributed fiber optic sensors with fiber Bragg gratings, which enables accurately the localization of a structural damage during the long-term monitoring of fully distributed fiber optic sensors. Moreover, the coupling mechanism of the reflected light from fiber grating and excited Brillouin scattering light is illustrated. Further, it is experimentally verified by locating the cracks of 2 m long reinforced concrete beams. The experimental results show that this proposed method is capable of monitoring the generation of the beam crack and further locating the crack on the concrete beam with an approximate error of 10 cm.

## 1. Introduction

Some long-distance structures such as pipelines, bridges, highways and railways play an important role in national economies and are prone to be damaged arising from the falling stone, sinkhole collapse and heavy loads. Under the action of these adverse loads, these structures are prone to long-term cumulative damage or sudden damage [1,2]. Many scholars have adopted different methods to monitor concrete cracks [3,4,5]. Deformation, strain and stress are the most important parameters for structural damage identification. In order to correctly measure and locate the structural deformation and stress, the sensors must be deformed in coordination with the structures. Fiber grating sensing technology and distributed optic fiber sensing technology [6,7,8] have been widely used in the engineering of structure monitoring [9,10,11,12,13,14,15,16] due to their high accuracy, strong resistance to electromagnetic interference, light weight, long distance and other advantages. Generally, the damage of the long-distance structure has the characteristics of wide distribution and strong randomness. Damage identification and localization are of great significance for long-term structural monitoring, providing the basis for maintaining the safety and stability of the structure. Brillouin optical time-domain analysis (BOTDA) technology is popularly and increasingly being utilized to measure strains in concrete structures. Feng [13] enhanced the measured strain by using the rubber gasket. Fan [16] developed a new method to investigate the concrete deterioration under steel corrosion with the use of a distributed fiber optic sensor. Mao [17] monitored concrete cracking under practical conditions by using BOTDA and fiber Bragg gratings (FBG). The optical time-domain reflectometer (OTDR) is a common piece of metrology equipment for evaluating optical fiber uniformity. An OTDR trace shows not only the fiber attenuation but also the magnitude and location of any losses and reflections along the fiber’s length. Crunelle [18] proposed a novel interrogating device that allows a very large number of sensing points to be detected and located. Yang [19] proposed a method to detect the damage of high-voltage transmission lines by OTDR. There are other methods to locate the cracks based on FBG. Mao [20] measured the cracks in reinforced concrete by FBG. Inners [21] monitored the cracks by acoustic emission. Among the existing damage localization methods, the widely used ones are fiber Bragg grating (FBG) localized methods [18,21,22] and fully distributed positioning methods of optical sensing fiber [13,16,17,19,20,22,23,24]. The above-mentioned methods either have a limited measurement range or many sensors to locate the damage sites and fail to achieve a precise spatial positioning for a long-distance distributed sensor.

In order to solve the pinpointing of crack monitoring for fully distributed fiber optic sensors, this paper proposes a positioning method of a combination of FBG and a fully distributed fiber optic sensor. The FBG and fully distributed fiber optic sensor are in a single sensing fiber. FBG was used as a localization indicator to eliminate the positioning error induced by the fiber length, refractive index, strain and ambient temperature. Meanwhile, a 2 m long concrete beam crack experiment was performed to validate its positioning accuracy.

## 2. Analysis of Brillouin-Scattering-Based Sensing Principle and Time-Domain Localization Theory

A Brillouin-scattering-based fully distributed fiber optic sensor enables the measurement of temperature and strain along the fiber based on Brillouin frequency shift:(1)$$v_{B}\left(\varepsilon ,T\right)=\frac{2v_{0}}{c}n(\varepsilon ,T)\sqrt{\frac{E(\varepsilon ,T)\left[1-k(\varepsilon ,T)\right]}{\left[1+k(\varepsilon ,T)\right]\left[1-2k(\varepsilon ,T)\right]\rho (\varepsilon ,T)}}$$
where n is the refractive index of the fiber, $v_{B}(\varepsilon ,T)$ is the Brillouin frequency shift, E is the Young’s modulus of the fiber, ρ is the fiber density and k is the Poisson’s ratio of the fiber. The above-mentioned parameters are related to temperature and strain. The relationship between strain and temperature and Brillouin frequency shift can be expressed as:(2)$$v_{B}\left(\varepsilon ,T\right)=v_{B}(0)+C_{T}(T-T_{0})+C_{\varepsilon }\varepsilon$$
where $v_{B}(0)$ is the initial strain, $C_{\varepsilon }$ is a strain sensitivity factor, typically 0.048 MHz/µε, $C_{T}$ is a temperature sensitivity coefficient, typically 1.07 MHz/°C, and $T-T_{0}$ is the temperature difference (°C). From Equation (2), the variation of temperature and strain is reflected by the magnitude of the Brillouin frequency shift [13].

Meanwhile, the distance from any point in the fiber to the detection light source can be measured by the of OTDR technology and get a spatial localization [6]:(3)$$z=\frac{ct}{2n}$$
where c is the propagation speed of light waves in optical fiber (m/s), n is the refractive index of optical fiber and t is the time of light propagation (s). Equation (3) can be differentiated as:
(4)$$dz=-\frac{ct}{2n^{2}}dn=-\frac{L}{n^{2}}dn$$

Equation (4) shows that the spatial position of the sensing fiber is influenced by three factors: (1) the refractive index of the fiber n; (2) the length of the fiber L; and (3) Δn induced by the strain and ambient temperature. The positioning error of spatial localization for a one-kilometer-long fiber is calculated to be 1.5 m [25] using the parameters of fiber refractive index variation of 0.811 × 10^−5^/°C and −0.1649 × 10^−6^/με, which arises from a ambient temperature change of 60 °C and a 1000 με deformation of sensing fiber Accordingly, the positioning error is much more greater for up to hundreds of kilometers of fully distributed fiber sensor. In addition, factors such as the performance of the fully distributed sensing test instrument itself [17,26] generates a positioning error. Therefore, it is essential to improve the localization accuracy of fully distributed sensors. In order to decline or even eliminate the positioning error of a long-distance distributed sensor, a combined FBG and BOTDA sensing system was proposed. FBG is used as a localization indicator in fully distributed fiber sensor [27,28]. Hence, the location of FBG transforms a long-distance distributed sensor into a short-distance distributed sensor.

## 3. Experiment

### 3.1. Coupling Test of Fiber Grating and Brillouin Scattering

In the combined FBG and BOTDA sensing system, the authors used FBG as a localization indicator in fully distributed fiber sensing [27,28]. To explore the coupling mechanism of fiber grating and excited Brillouin scattering, the optical path (Figure 1) was set to detect the pump and probe end light source spectra of an NBX-6040A instrument. Meanwhile, the coupling spectra between the pump and probe light sources and three purchased FBGs were measured by AQ6317 optical spectrum analyzer (OSA). Their performance parameters are shown in Table 1. In addition, the Brillouin optical power spectrum of six FBGs was measured (Figure 2). The wavelength signal of FBG was recorded by OSA with a wavelength repeatability of 10 pm.

### 3.2. Concrete Crack Location Experiment

The sensing fiber used in the experiment was a single-mode silicon fiber with a white protective sleeve, 900 μm in diameter. Two FBGs were used as positioning tools, named FBG1 and FBG2. Their performance parameters are shown in Table 2.

To detect cracks of different widths, there were two pasting methods used on the surface of the concrete beam. The 1.5 m length sensing fiber was fully pasted along the concrete beam, named full adhesive type. The other method is point type, which had a 1.5 m distributed sensing fiber glued on the beam with a 10 cm interval. The fully pasted sensing fiber detects small cracks. On the contrary, the point-glued method is used for detecting a wide crack. The first three segments were fully adhesive optical fiber, and the last three segments were point-paste type optical fiber, respectively labeled as 1, 2, 3, 4, 5 and 6 (Figure 3). The location of the sensing fiber and FBG is shown in Figure 4; they were all pasted with two-component epoxy adhesive. In the experiment, the concrete beam was fixed on a steel frame, and bolts and channels were used to fix the ends of the concrete beam. Meanwhile, a QLD32 screw jack was used to load step-by-step, and the step load was recorded through the LH-S10-2T pressure transducer (Figure 5). The applied load was from 0 to 8 kN. The location and time of crack were simultaneously observed by the naked eye. The fiber strain was recorded by NEUBREX’s NBX-6040A high-precision analyzer with a spatial resolution of 10 cm. Its sampling rate and interval was 2 × 10^15^ and 5 cm, respectively.

## 4. Results and Discussion

### 4.1. Coupling Mechanism of FBG and Brillouin Scattering

Figure 6 and Figure 7 are pump and probe light source spectra, respectively. It is shown that the wavelength range of pump and probe light sources are from 1460 to 1600 nm. The central wavelength of pulsed pump light and continuous probe light is 1548.52 nm and 1548.61 nm, respectively.

To explore the coupling characteristics of Brillouin scattering and FBG, their coupling spectra were measured by AQ6317 spectrometer. As shown in Figure 8a, it was caused by the superposition of the reflection of the pump light through FBG and the transmission of the probe light through FBG. The transmission spectrum of FBG was formed due to higher-energy pump light, shown in Figure 8b. Subsequently, a AQ6317 spectrometer was used to test the spectrum of six FBGs in the wavelength range of 1460~1620 nm. Figure 9 is the coupling spectra between FBGs, probe and pump light. FBGs in the wavelength range of 1520~1580 nm have reflective spectra, while the other FBGs have transmission spectra (Figure 9) generated by the superposition of the two beams of light from the pump and probe end passing through FBGs. When the pump light power is higher than the probe light, FBGs have a reflection spectrum; otherwise, they have the transmission spectrum. Therefore, the light power peak of FBGs is generated by FBG reflection. When FBGs and fully distributed sensing fibers are measured in a single line, a light power peak occurs.

### 4.2. Concrete Crack Localization

Figure 10 shows the excited Brillouin scattering power distribution of a distributed sensing fiber sensor. The rectangular shading is the power reflection spectrum of FBG_1_ and FBG_2_. Their spatial locations along the sensing fiber were 3.131 m and 70.585 m, respectively. The actual distance from FBG_1_ to the beam end was 4.29 m. Therefore, the spatial locations of the three sections of the fully adhesive fiber were 7.421 m, 13.421 m and 16.421 m, respectively; the actual distance from FBG_2_ to the starting position of the concrete beam was 4.41 m, and the spatial positions of the three sections of point-sticky fiber were 74.995 m, 80.995 m and 83. 995 m, respectively. In addition, the graph also shows the power fluctuations along the fiber caused by fiber bends and flange connections.

Figure 11 shows the strain distribution along the sensing fiber without applied load. It is shown in Figure 11 that six fiber segments were initially prestressed. Due to the limitation of spatial resolution, the location of the beam end failed to be accurately determined. The enlarged graph (inset in Figure 11) illustrates the distributed strain of the first fully adhesive fiber segment. The red ball label is the location of the beam end (7.187 m) from the distributed strain curve, while its actual location was 7.421 m. Thus, the positioning error of the beam end was 0.234 m. However, the length between the beam end and FBG_1_ (3.131 m) was 4.29 m, and hence the location of beam end (7.421 m) was precisely calculated using the FBG_1_ spatial site (3.131 m). It is found that a positioning error is consequently eliminated using FBG_1_ as a localization indicator.

The cracks gradually generated with an increasing load. Eventually, there were nine transverse cracks successively observed by naked eye. An 8 kN load was ultimately applied to achieve differential concrete cracks. When the load was 4.5 kN, the ①, ②, ③, ④ and ⑤ cracks occurred and, subsequently, the 6 and ⑦ cracks at 5 kN and the ⑧ and ⑨ cracks at 8 kN, as shown in Figure 12.

Figure 13 shows the Brillouin frequency shift distribution along the sensing fiber. Figure 13b–d show an enlarged Brillouin frequency shift distribution of the three segments of the fully pasted fiber sensor, respectively. The green rectangular area is the beam length, and sharp frequency shift peaks appear at the 1, 2 and 3 fiber segments. The first frequency shift peak appeared when the beam was loaded to 4.5 kN, subsequently, the second sharp peak was generated at the 7 kN load and, ultimately, the third sharp peak at 7.5 kN load. The same experimental findings are observed in Figure 13c,d. Hence, the Brillouin frequency shift spectrum reflected the number of cracks well.

Figure 14 shows the crack location of the full-bonded sensing fiber and the actual location of the crack. In Figure 14, the black line represents the actual location of a crack, the red, blue and yellow lines represent correspondingly positioning sites of these cracks and their generated order. The crack locations are mainly concentrated in the rectangular shaded area, and the ①, ②, ③, ④ and ⑦ cracks were precisely located. At the same time, the identical cracks (Table 3) are identified by the 1, 2 and 3 segments of the sensing fiber. As a result, their positioning errors of corresponding cracks were approximately 10 cm. The error is perhaps attributable to: (i) a lower spatial resolution; (ii) a positioning error of FBG is 2–3 cm [27,28]; (iii) an error induced by the actual length measurement of used optical fiber. In addition, the ⑤, ⑧ and ⑨ cracks were not identified either due to fiber debonding or the closely positioned cracks being unidentified due to a lower spatial resolution of the BOTDA instrument.

Figure 15 shows the Brillouin frequency shift spectrum along the point-glued fiber and Figure 15b–d are the enlarged Brillouin frequency shifts of 4, 5 and 6 point-glued fiber segments, respectively. Compared with the fully bonded fiber sensor, the Brillouin frequency shift of the 4, 5 and 6 segments increased as the load increased, step-by-step, while sharp Brillouin frequency shift peaks failed to occur. The point-bonding method contributed to the experimental phenomena. The sensing fiber was uniformly stressed with the increasing load, consequently, the Brillouin frequency shift increased and uniformly distributed along the sensing fiber. Therefore, each crack cannot be identified when the generated cracks (Figure 15) are basically located in the middle part of the beam (green shading) using the point-glued method. 

When the concrete beam was loaded to 4 kN, the ①, ②, ③, ④ and ⑤ cracks appeared, the Brillouin frequency shift of the 4, 5 and 6 segments of the sensing fiber increased, and the wave trough of Brillouin frequency shift spectrum simultaneously appeared. The wave trough disappeared at a 5 kN load. The location of the trough at different segment of sensing fiber was 75.821 m, 80.184 m and 84.856 m, respectively. Due to the generation of the ⑦ crack located between the ③ and ④ cracks, the location of the trough was the ⑦ crack.

In order to verify whether the location of the trough was the ⑦ crack, the hot water bag was placed at the ⑦ crack. As observed in Figure 16, Brillouin frequency shift at the location of the wave trough increased significantly with an increasing temperature. Therefore, it is inferred that the wave trough was experimentally verified to be the location of the crack. The location of the ⑦ crack was 75.821 m, 80.195 m and 84.795 m on the 4, 5 and 6 segments of the sensing fiber, respectively. The location of the beam end was correspondingly calculated as 74.995 m, 80.995 m and 83.995 m. Thus, the distance between the ⑦ crack to the beam end is calculated to 0.826 m, 0.811 m and 0.861 m. Accordingly, the identified ⑦ crack location is basically identical to its actual location (Figure 17).

## 5. Conclusions

In this paper, the FBG-based pinpointing method was proposed and the coupling characteristics between the reflected light from FBG and Brillouin scattering light was illustrated. It was utilized to locate cracks in a 2 m long concrete beam and then the feasibility of the locating method was verified. The following main conclusions are:

(1) FBG was used as a location indicator in a pinpointing method for a fully distributed fiber optic sensor. The experimental results indicate that the appearance of power peaks in the excited Brillouin power spectrum is caused by the reflection of FBG in the wavelength range of 1520~1580 nm. The coupling mechanism between the reflection of FBG and the Brillouin scattering light is the superposition of the pump light reflection from FBG and the probe light transmission through FBG.

(2) The positioning experiment of cracks using FBG-based pinpointing method showed that the crack generation and propagation process of a concrete beam can not only be accurately recorded but can also be pinpointed with a 10 cm error. This work provides a method for early warning and location of structural damage.

## Figures and Tables

**Figure 1 micromachines-12-01479-f001:**
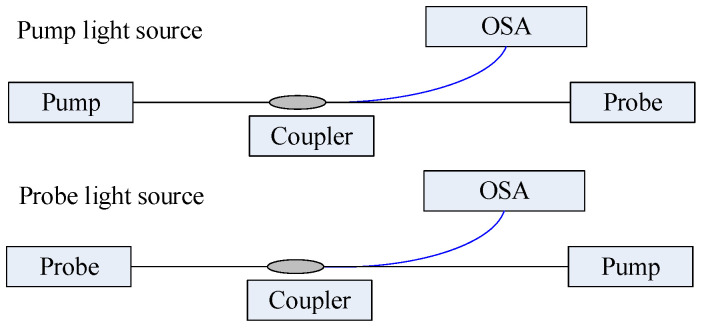
Schematic diagram of the optical path of pump and probe light sources for spectral testing.

**Figure 2 micromachines-12-01479-f002:**
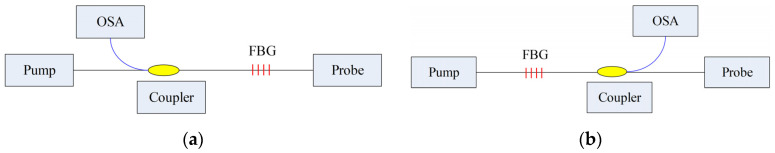
Spectral test optical path diagram. (**a**) Pump; (**b**) probe.

**Figure 3 micromachines-12-01479-f003:**
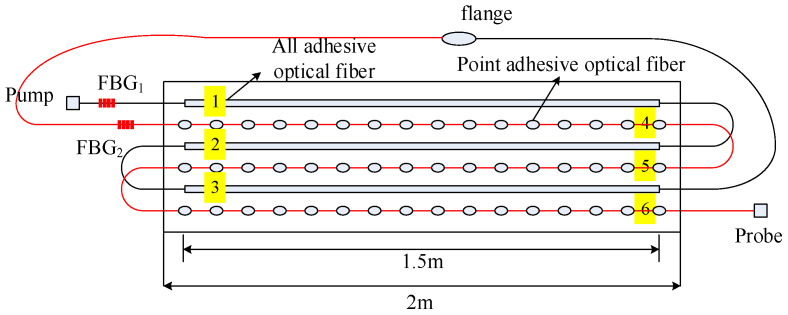
Diagram of fiber optic paste method.

**Figure 4 micromachines-12-01479-f004:**
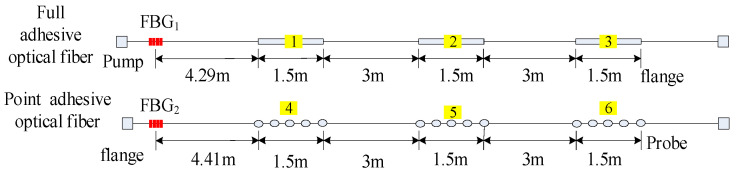
Relative spatial position of sensing fiber and fiber grating.

**Figure 5 micromachines-12-01479-f005:**
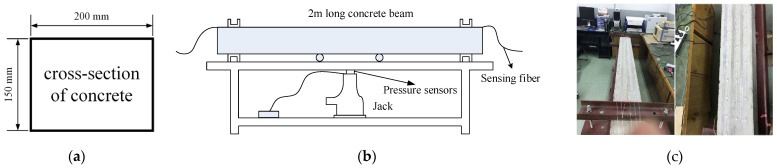
Schematic diagram of experimental device. (**a**) Cross-section dimensions of the beam; (**b**) schematic diagram of experiment; (**c**) experimental diagram.

**Figure 6 micromachines-12-01479-f006:**
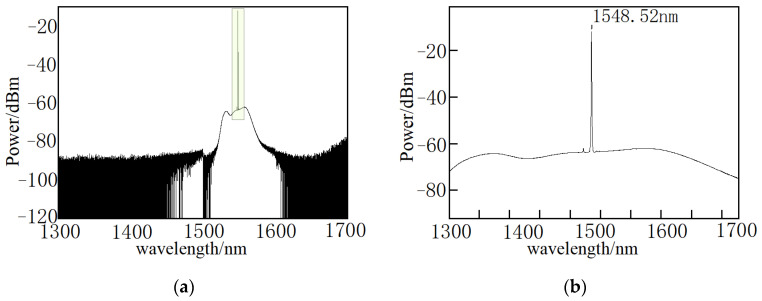
Pump light spectrum. (**a**) overall spectrum; (**b**) central wavelength of pump light.

**Figure 7 micromachines-12-01479-f007:**
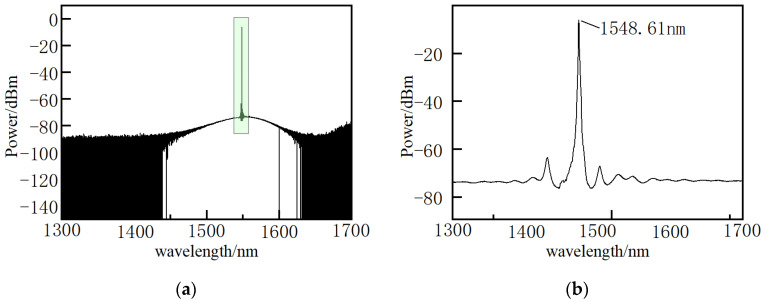
Probe light spectrum. (**a**) Overall spectrum; (**b**) central wavelength of probe light.

**Figure 8 micromachines-12-01479-f008:**
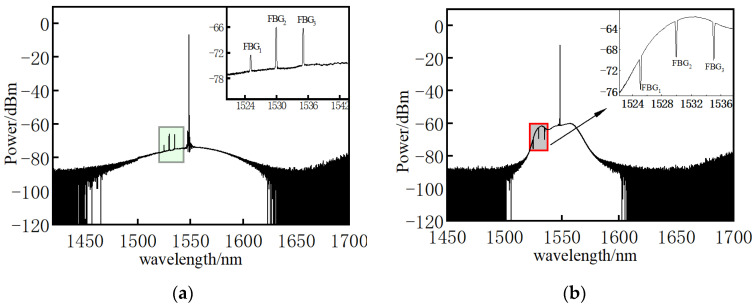
Interactive spectra. (**a**) Probe and FBG; (**b**) pump and FBG.

**Figure 9 micromachines-12-01479-f009:**
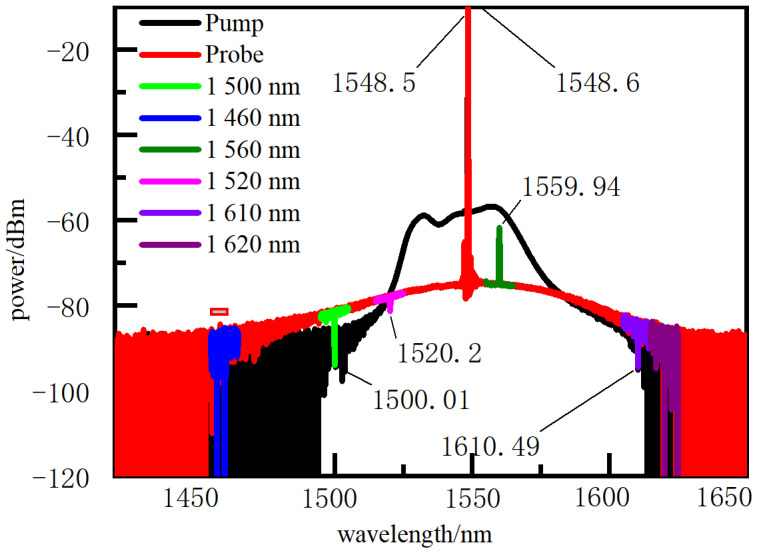
FBGs and BOTDA double-ended spectra.

**Figure 10 micromachines-12-01479-f010:**
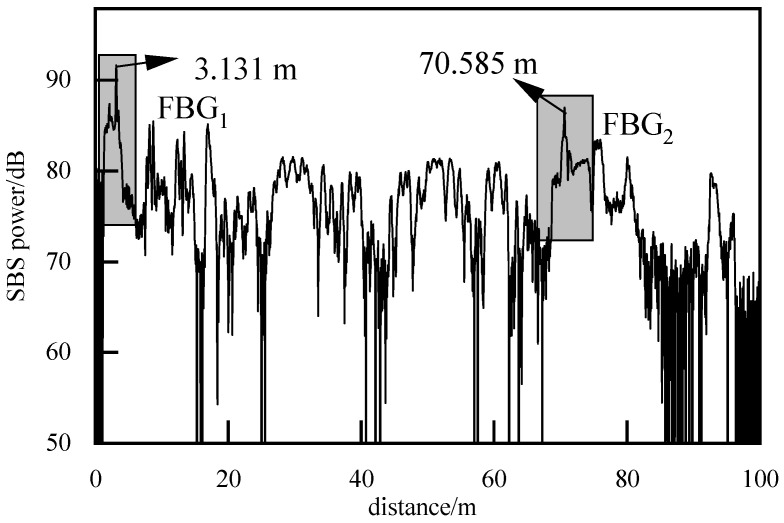
Power spectrum of stimulated Brillouin scattering.

**Figure 11 micromachines-12-01479-f011:**
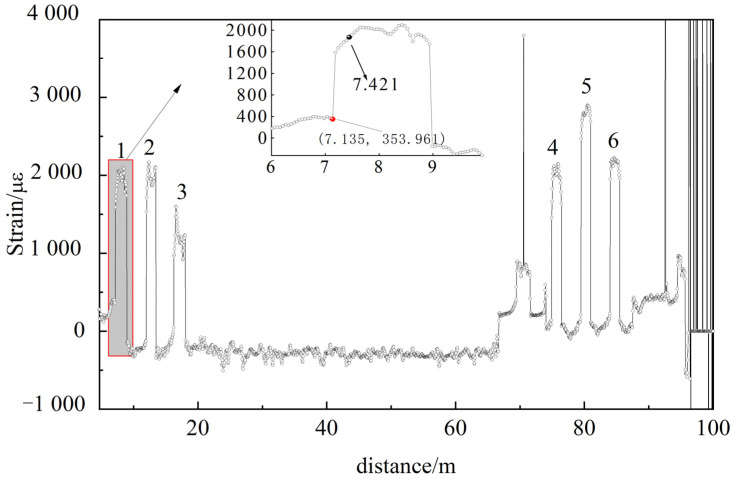
Distributed strain along the sensing fiber without applied load.

**Figure 12 micromachines-12-01479-f012:**
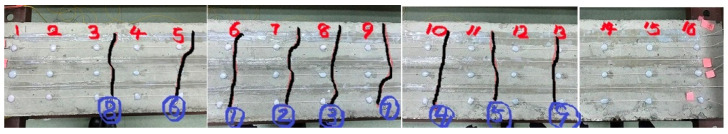
Crack distribution along the concrete beam.

**Figure 13 micromachines-12-01479-f013:**
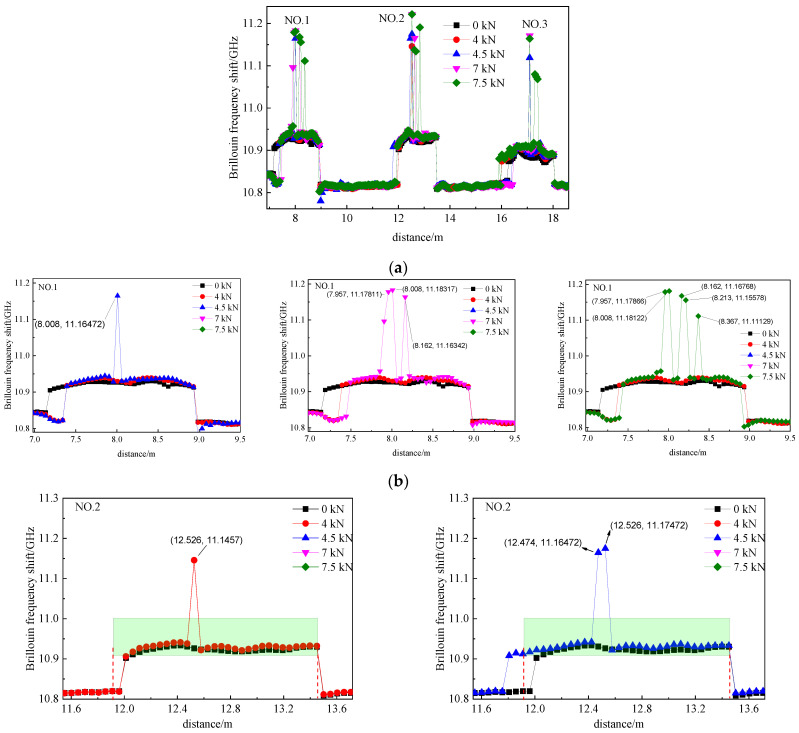
Brillouin frequency shift along the fully bonded optical fiber. (**a**) Overall view of Brillouin frequency shift of 3 segments of fiber; (**b**) enlarged view of the first segment pasted optical fiber; (**c**) enlarged view of the second segment pasted optical fiber; (**d**) enlarged view of the third segment pasted optical fiber.

**Figure 14 micromachines-12-01479-f014:**
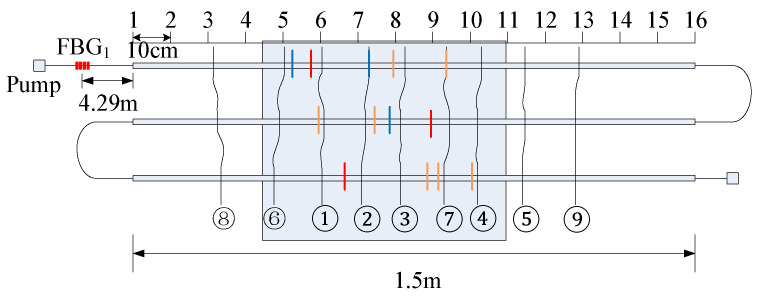
Position of concrete cracks using the full-pasted fiber sensor.

**Figure 15 micromachines-12-01479-f015:**
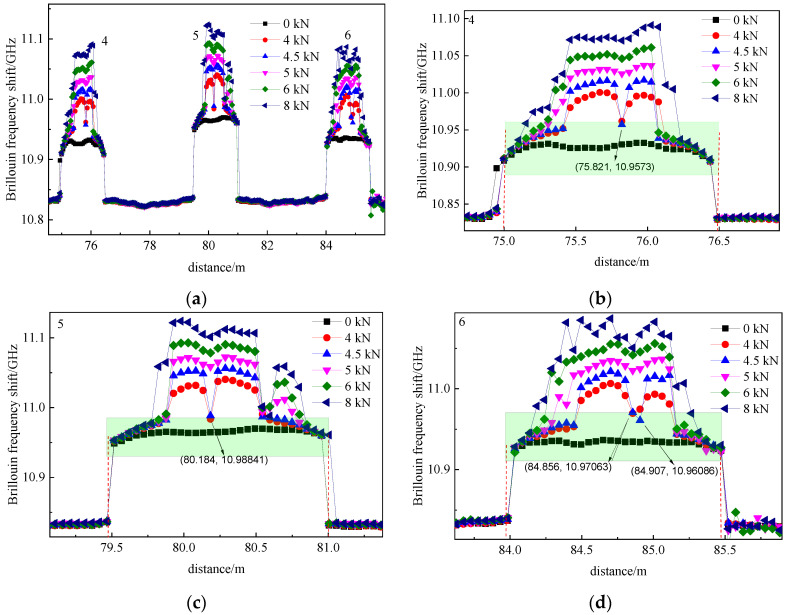
Brillouin frequency shift along the pointed-bonded optical fiber. (**a**) Overall view; (**b**) the fourth segment pasted optical fiber; (**c**) the fifth segment pasted optical fiber; (**d**) the sixth segment pasted optical fiber.

**Figure 16 micromachines-12-01479-f016:**
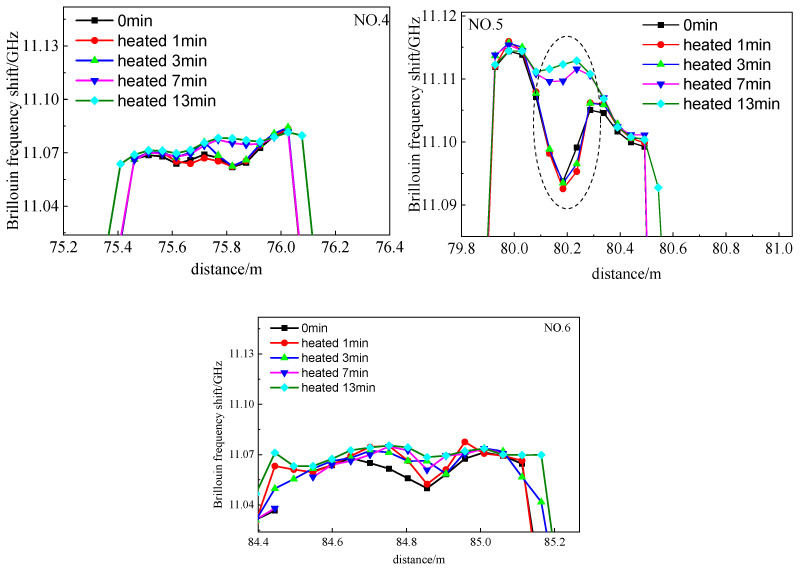
Frequency spectrum of point-glued sensing fiber under the varied temperature.

**Figure 17 micromachines-12-01479-f017:**
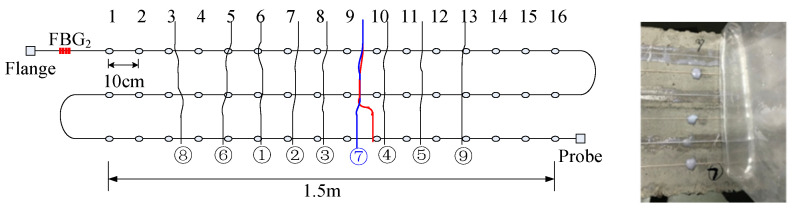
Schematic diagram of crack location of point bonded optical fiber (blue: actual location, red: FBG locating cracks).

**Table 1 micromachines-12-01479-t001:** Properties of FBGs along the sensing fiber.

Number of FBG	Performance Parameters		
	Wavelength/nm	Bandwidth/nm	Reflectance %
FBG1	1525.03	0.23	84.54
FBG2	1529.88	0.21	82.59
FBG3	1535.01	0.21	84

**Table 2 micromachines-12-01479-t002:** Performance parameters of FBG sensor.

Number of FBG	Performance Parameters		
	Wavelength/nm	Bandwidth/nm	Reflectance %
FBG1	1544.74	0.22	73.39
FBG2	1529.91	0.23	91.54

**Table 3 micromachines-12-01479-t003:** Performance parameters of FBG sensor.

Segment	Distance/m	Identification Length/m	Crack Number	Actual Length /m	Positioning Error/m
1	7.957	0.536	6	0.4	0.136
	8.008	0.587	1	0.5	0.087
	8.162	0.741	2	0.61	0.131
	8.213	0.792	3	0.7	0.092
	8.367	0.946	7	0.82	0.126
2	12.834	0.587	1	0.5	0.095
	12.680	0.741	2	0.63	0.093
	12.628	0.793	3	0.7	0.111
	12.526	0.895	7	0.83	0.065
3	17.094	0.673	2	0.61	0.063
	17.300	0.879	7	0.83	0.049
	17.351	0.93	7	0.83	0.100
	17.402	0.981	4 or 9	0.92 or 1.03	0.061 or 0.049

## Data Availability

The data presented in this study are available from the corresponding author, [J.L.], upon reasonable request.

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
