# Peer review of "A Combined Positioning Method Used for Identification of Concrete Cracks"

_micromachines, 2021, doi:10.3390/mi12121479_

Round 1

Reviewer 1 Report

In the manuscript, titled A combined positioning method used for identification of concrete cracks, the authors used FBGs to realize a more accurate cracks location. This topic may raise the interests of the community. And the experimental results clearly demonstrate determination of location. However, the manuscript cannot be published at its current state, the authors need to address the following aspects:

  1. The authors need to rephrase many sentences to make them concise.
  2. 4 mat be wrong, I think it should be -L/n dn.
  3. Can the author provide any experiment result to validate Eq 4?

Author Response

Our response:

We have carefully checked this manuscript and improved the language.

Our response:

Thank you for pointing out this mistake, we have corrected it as “ ”.

Among the equation, the denominator is n 2 instead of n. The right part of Equation (3)  has a single Variable n. Taking the derivative of , it is consequently .

Our response:

Equation (4) shows that the spatial position of the sensing fiber is influenced by three factors. (1) the refractive index of the fiber; (2) the length of the fiber; and (3) the strain of the fiber and the change in the temperature of the environment, the influence of . The authors intend to indicate the spatial location along the sensing fiber is theoretically varied with the , , induced by the strain and temperature. Therefore, the experiment result is inessential to investigate the pinpoint the concrete crack.

Thank you for the comments provided here. Your comments helped us to greatly improve our manuscript and are appreciated.

Reviewer 2 Report

Comments

  1. Introduction: It is suggested to add additional text to this section explaining the need for a coupling mechanism to monitor concrete structure.  The motivation for why this civil engineering structural health monitoring study, specifically for concrete beams, is not full presented in this section. For example, what previous studies for concrete structures utilizing FBG, distributed FOS have been performed in the literature?  MDPI Sensors Journal (and other journals), for example, has articles covering civil engineering fiber optic sensors applications that can be included in the introduction to add context to this work.
  2.  The novelty/significance is not being fully highlighted for this work.  A thorough literature review of the state of the art for Brillouin scattering and FBG for civil engineering structures would help build arguments for this study.
    3. Section 3.2, Figure 1, and Figure 2: Additional text explaining the motivation for using the pasting methods is needed in this section.
    4. Section 3.2:  Were the FBG sensors acquired from a vendor or were they manufactured by the authors?  
    4. What type of paste/adhesives were used? What was the reason for using the particular pastes/adhesives?
    5. Also, what is the logic behind loading to 8 kN. Were there any mechanical testing theories applied or calculations performed loading the concrete beam? This should be thoroughly explained with statements added to the manuscript.
    6. For Figure 5, the cross-section dimensions of the beam should be added.

Minor Comments:
Figure 1: Define "OSA" in Section 3.1
Table 1:  Add a statement to give reviewers the significance of the Reflection values.

This work has merit and can be re-considered for publication after addressing comments.

Author Response

Dear Editor, Dear reviewers

Thank you very much for taking your time to review this manuscript. We truly appreciate all your comments and suggestions. Based on the comments provided in your letter, we have uploaded this letter for reviewer response. Accordingly, we uploaded the revised manuscript with all the corrections, the file is MS Word document in "track changes" mode.

Appending this letter is our point-by-point response to the comments raised by the reviewers. Our responses are given directly afterwards for each comment.

We would also like to thank you for allowing us to resubmit a revised manuscript.

We hope that the revised manuscript is accepted for publication in the Micromachines.

Reviewer 2:

Our response:

We have added some reference in Introduction to explain the need for concrete monitoring.

Meanwhile, the coupling mechanism between FBG and fully distributed sensor has been added in the introduction.

Our response:

We have added some previous work of concrete monitoring in the Introduction of revised manuscript and highlight the motivation in this study.

Our response:

To detect cracks of different widths, there are two pasting methods used on the surface of the concrete beam. The fully pasted sensing fiber detect a small crack. On the contrary, the point-glued method is used for detecting a wide crack.

We have added the above discussion to Section 3.2.

Our response:

The FBG were acquired from a vendor. We have added the description to Section 3.1 in revised manuscript.

Our response:

We mainly used a two-component epoxy adhesive for pasting the sensing fiber to achieve a high bonding strength between a distributed sensor and concrete in the experiment. The description of glue is supplemented in Section 3.2.

Our response:

To ensure the generation of concrete cracks and prevent the break of distributed sensor, an 8kN load was ultimately applied for achieve concrete cracks with differential widths.

We added some description in Section 4.2.

Our response:

We have added the cross-section dimensions of the beam in Figure 5.

We have added the description of OSA in Section 3.1.

Thank you for the comments provided here. Your comments helped us to greatly improve our manuscript and are appreciated.
